# The COVID-19 Health Protocol among University Students: Case Studies in Three Cities in Indonesia

**DOI:** 10.3390/ijerph191710630

**Published:** 2022-08-26

**Authors:** D. Daniel, Arif Kurniawan, Ajeng Rahastri Indah Pinawati, Morrin Choirunnisa Thohira, Md Annaduzzaman

**Affiliations:** 1Department of Health Behavior, Environment, and Social Medicine, Faculty of Medicine, Public Health and Nursing, Universitas Gadjah Mada, Sleman 55281, Indonesia; 2Department of Environmental Science, The Graduate School, Universitas Gadjah Mada, Sleman 55281, Indonesia; 3Department of Public Health, The Graduate School, Universitas Gadjah Mada, Sleman 55281, Indonesia; 4Sanitary Engineering Section, Faculty of Civil Engineering and Geosciences, Delft University of Technology, 2600 Delft, The Netherlands

**Keywords:** COVID-19, health protocol, Indonesia, university students, RANAS framework

## Abstract

The COVID-19 pandemic has caused lifestyle changes for everyone and led to the practice of regulated health protocols for preventing the spreading or severity of the COVID-19 pandemic. This study examines the differences in health protocols and health practices among university students. The designed online survey was conducted among 292 university students in three cities in Indonesia, i.e., Yogyakarta, Semarang, and Surakarta. A forced-entry multivariate regression was conducted using all RANAS (risk, attitude, norms, ability-self-regulation) sub-factors as independent variables and health protocol obtained from PCA as the dependent variable. The results showed that the students’ health protocol and health practices were practiced with varying frequency. A face mask covering the chin and nose was the most practiced health protocol, while reducing mobilization by maintaining distance was the most violated health protocol among students. We also found that four health protocol practices are highly correlated, i.e., handwashing in public spaces, physical distancing, frequency of using the mask, and avoiding crowded places. In addition, three significant psychological factors were identified, which were positively associated with the student’s health protocol practice, i.e., belief about time (attitude) (OR: 0.119; CI: −0.054–0.136; *p* ≤ 0.05), personal norm (norm) (OR: 0.232; CI: 0.149–0.539; *p* ≤ 0.01), and action control (self-regulation) (OR: 0.173; CI: 0.046–0.427; *p* ≤ 0.05), where the personal norm is the most significant one. Finally, to minimize the COVID-19 transmission among students, especially when they back to onsite learning, it was important to create students’ sense of ethical self-obligation to follow and practice standard health hygiene correctly and regularly.

## 1. Introduction

The novel coronavirus disease (COVID-19) is one of the numerous communicable diseases that spread rapidly to the human body. People who are infected with COVID-19 will experience mild to moderate symptoms; however, those who have the underlying medical condition are likely to become more severely ill, which can lead to death [1]. Due to its wide and rapid spread, COVID-19 has become a pandemic since March 2020 [2]. WHO (February 2022) reported that around four hundred million people were identified with COVID-19 cases and 5 million deaths [3].

The total cases in Indonesia on 4 February 2022 were about 4.5 million confirmed cases and 140 thousand deaths [3]. The only method to reduce the transmission of COVID-19 is by implementing health protocols. These health protocols consist of wearing a face mask, washing hands using soap, maintaining physical distancing, avoiding gathering or crowds, and reducing mobilization [4,5]. Oh et al., 2020 [6], indicated that proper health protocol practices can minimize COVID-19 transmission. However, the lack of awareness and proper hygiene or health protocol practice leads to a high infection and mortality rate in Indonesia [7].

Most studies on COVID-19 health protocols use general citizens as the respondents, e.g., residents of a country by region [8,9,10], while a small portion focuses on health workers and older adults [11,12]. However, few studies are focusing on COVID-19 health protocol in students [13,14]. Despite public health scholars arguing that susceptibility to infection in the young population is relatively smaller compared to the old population [15], the young population is considered strong COVID-19 virus carriers able to transmit to elderly people. Therefore, it is still important to understand health protocol practices among this group. That is because university students are among the most strongly affected by COVID-19, in context of, e.g., uncertainty regarding academic success, future careers, and social life during college, among other concerns [16]. Moreover, a previous study among university students found that students often ignore health protocols, an unsatisfactory level of knowledge, and an attitude regarding COVID-19 [14,17]. Students can easily spread the virus into the community, due to the lack of understanding of the long-term effects caused by COVID-19 in the community [18]. This underlines the high risk of COVID-19 transmission among university students.

The practicing of health protocols is influenced by people’s perceptions, i.e., psychological factors. Some psychological frameworks or theories have been used to analyze the practice of health protocols, e.g., the health belief model, the reasoned action approach, and the risk, attitude, norms, ability, and self-regulation (RANAS) framework [8,9,19]. Those studies found several significant psychological factors related to the practice of health protocol, e.g., the social norm in the community, injunctive norms, intention, and perceived self-efficacy. By understanding and targeting those significant factors in the implementation, it is believed that people will change their behavior, i.e., practice regularly and appropriately the health protocol [20].

However, a very limited study analyzes the underlying psychological factors related to the COVID-19 health protocols among university students. For example, Sharma et al. [13] used Multi-Theory Model (MTM) to analyze health protocols in students but only for hand washing practices. However, we argue that analyzing other health protocols and practices among university students, including underlying psychological factors, could provide important insight into the prevention or spread of COVID-19 within different age groups and/or communities. Therefore, this study aims to identify and fill the knowledge gap about the COVID-19 health protocol among university students. More specifically, the research questions are: (1) “How is the practice of health protocols among university students in Indonesia?” and (2) “What are the psychological factors that influence those health protocols?”.

The online survey data of university students from three cities in Indonesia were collected for this study: Yogyakarta, Semarang, and Surakarta. RANAS psychological framework was followed to understand underlying psychological factors of the COVID-19 health protocol practice. The RANAS framework consists of psychological factors related to risk, attitude, norm, ability, and self-regulation that potentially affect personal behavior [9,20]. The changes in the health protocols practice were also investigated before COVID-19 entered Indonesia, at a later point when the cases were still low, and at the peak phase of the pandemic in 2021, which was caused by the Delta variant [21], even though this study is more focuses on the peak phase, i.e., when the data was collected. This may help us understand the dynamics of health protocol practices among university students. The data were analyzed statistically, especially to find out significant psychological factors associated with the health protocols. Understanding the underlying psychological factors behind the behavior is necessary to design relevant strategies to prevent COVID-19 transmission among university students.

## 2. Materials and Methods

### 2.1. Data Collection

The study used a cross-sectional design. The purpose of this study was to gain an overview of changes in university students’ behavior in implementing health protocol (wearing masks, maintaining distance, washing hands, staying away from crowds, and limiting mobility) related to the COVID-19 pandemic in 3-time phases (before the pandemic, i.e., before January 2020; during low cases, i.e., around April 2021; and the peak of the pandemic, i.e., July 2021), which are shown in Figure 1.

The data was collected by a self-reported online questionnaire (Google form) for the period of July 2021 to February 2022. It is important to note that the data collection was conducted only during July 2021–February 2022, but we asked respondents to recall their behavior for the first phase, i.e., before the pandemic, and the second phase, i.e., low cases. Google form is one of the instruments available in the Google workspace. It is used for creating online questionnaires that can be easily published and analyzed in the real time [22]. The most important benefit of using Google Form is the ability to easily obtain people’s opinion remotely [23]. In this research, we created questions about the respondent’s background (number of questions = 8), the implementation of health protocol (n = 11), and RANAS (Risks (n = 7), Attitudes (n = 7), Norms (n = 3), Abilities (n = 3), and Self-regulation (n = 7)). The participants in this study were students in three major cities in Indonesia: Yogyakarta, Semarang, and Surakarta. These three cities were chosen because they have a large number of universities and students. Yogyakarta has 135 universities with 1.1 million students, Semarang has 64 universities with two hundred thousand students, and Surakarta has 42 universities with almost a hundred thousand students [24,25]. These areas have more than 130 universities and a large number of students. A number of 292 students participated voluntarily in this online survey.

Participation in this research was on a voluntary basis. All stages of the research went through ethical clearance from the Ethics Commission of the Faculty of Medicine, Public Health, and Nursing, Universitas Gadjah Mada. Confidentiality of information and participant rights and obligations were included in the informed consent.

### 2.2. RANAS (Risk, Attitude, Norm, Ability, and Self-Regulation) Psychological Framework

The RANAS model is a method used to design and evaluate behavior change in a population [20]. There are five main factors in this method, they are Risk, Attitude, Norm, Ability, and Self-regulation (RANAS). “Risk” is related to the understanding and awareness of health protocols. “Attitude” is related to students’ positive or negative feelings about the health protocols. “Norm” represents social pressure towards behaviors. “Ability” is related to student capability to commit and practice health protocols. Lastly, “Self-regulation” represents one’s plan to self-monitor health protocols [26]. RANAS framework inquires psychological information at the sub-factor level [20], see also Table 1. All scales used in the RANAS-related questions follow the RANAS guidelines [27]. Previous studies also used the RANAS framework model to find critical psychological factors that influence health protocols, for example, public perception, hand hygiene behavior [9], and mask-wearing [10].

### 2.3. The COVID-19 Health Protocols and Daily Healthy Practices

A method for increasing health protection from the coronavirus transmission is the implementation of health protocols. Individual health protocols consist of handwashing using soap, wearing a mask covering the nose and chin properly, maintaining physical distancing, avoiding crowds, and reducing mobilization [24]. In this study, some additional variables/measures related to daily health practices were included, i.e., covering mouth and nose when sneezing, the frequency of doing physical exercise, the frequency of vitamin or supplement consumption, and balanced nutrition. The measurement of the frequency of health protocols and daily healthy practices using Likert scales, which were divided further into five scales: never, very rarely, seldom, almost, and always, except reducing mobilization, where the score range was 1 (best practices) to 10 (worst practices), and physical exercise, where the range score was 1 (no physical exercise at all) to 9 (seven days a week).

### 2.4. Data Analysis

A total of 292 respondents filled out the questionnaire and the data were entered into Microsoft Excel for data cleaning. However, only 78 respondents answered the questions related to mobilization. Afterward, the data from Microsoft Excel was entered into IBM SPSS version 26 (IBM, New York, NY, USA) for statistical analyses [28]. Furthermore, since there is a difference in scale between health protocol and daily healthy practice variables, the scales of mobilization (10 scales) and physical exercise (9 scales) variables were divided into five scales to allow meaningful comparison between them.

PCA was used to create a latent variable and reduce the dimensionality of variables in the analysis. PCA yields a variable that represents the health protocols of respondents, which comprised some actions or practices, e.g., handwashing, physical distancing, and using a face mask. The PCA procedure was deemed acceptable when the minimum Kaiser–Meyer–Olkin (KMO) value was 0.5 [29] and the Cronbach’s α value was above 0.7 [30]. The first principal component was extracted assuming that it represents the health protocol practices of the respondents.

A forced-entry multivariate regression was conducted using all RANAS sub-factors as independent variables and the health protocol was obtained from PCA as the dependent variable. The health protocol was created using PCA from three main health protocol practices: handwashing in the public space, physical distancing, and the frequency of mask use. For the regression, we used the PCA of health protocol practices at the peak of the COVID-19 pandemic, i.e., July 2021.

## 3. Results

### 3.1. Characteristics of and Health Protocol Practiced

Among 292 respondents, 74% were female and 26% were male. The age group averaged was between 23 years old (SD = 4.10; range: 17–43). Furthermore, more than half of the respondents were studying in the city of Yogyakarta (58.6%), followed by Semarang (27.2%) and Surakarta (13.7%). In terms of residence, 53.8% lived with their family, 39.4% in boarding houses, and the remainder in dormitories, apartments, Islamic boarding schools, official residences, and barracks. There were 233 respondents (79.8%) who had never been infected with COVID-19, 59 respondents (20.2%) had been at least once, and 2 had been infected more than one time. Most of the respondents had experienced COVID-19 infections in the family (67.1%).

### 3.2. Description of the Health Protocol

As stated previously, the changes in health protocols were divided into three phases. The first phase was before the pandemic (before January 2020), the second one was when the cases initially decreased (April 2021), and the third phase was when the cases reached their peak in July 2021. Health protocols practiced and daily health practices by respondents are shown in Table 2. All of the mean scores of health protocols and daily health practices increased from the first phase (pre-pandemic) to the third phase (post-pandemic). In other words, the implementation of health protocols and daily healthy practices were being improved by the participants when the number of cases increased rapidly.

Physical exercise has the lowest average score in each phase while covering the mouth and nose during sneezing tends to have a high score. The pre-pandemic phase had the highest average score for practicing mouth and nose covering during sneezing. Importantly, the average mask-use frequency increased during moderate and high infection periods. Figure 2 shows that the mobilization variable was the most difficult variable to be implemented, followed by physical distancing. These two practices were also often violated in the health protocols.

### 3.3. Principal Component Analysis (PCA)

We first conducted PCA using three health protocol practices at the peak period of COVID-19 cases, i.e., July 2021: handwashing in public spaces, physical distancing, and the frequency of mask use. The KMO value was 0.676, which can explain 64% of the variance (n = 292). Furthermore, the Cronbach’s α value was 0.714.

PCA by adding an extra health protocol practice, i.e., avoiding crowded places was also included. The KMO and Cronbach’s α values were 0.723 and 0.725, respectively, which can explain 57% of the variance (n = 78). The result indicate that the level of students avoiding crowded places at the peak period of COVID-19 was related to the level of practices of the other three health protocols. However, due to a small sample size obtained for the second PCA, i.e., not enough samples for the regression analysis, we decided to use the principal component of the first PCA in the regression analysis.

### 3.4. Regression Analysis

The results of the regression analysis are shown in Table 3. Respondent’s cities are included as control variables because we found that the PCA scores of health protocol among students from Semarang were significantly lower compared to other cities (H(2) = 8.58, *p* < 0.05). The results showed three significant psychological sub-factors related to the students’ health protocol, which were *belief about time* (attitude) (OR: 0.119; CI: −0.054–0.136; *p* ≤ 0.05), *personal norm* (norm) (OR: 0.232; CI: 0.149–0.539; *p* ≤ 0.01), and *action control* (self-regulation) (OR: 0.173; CI: 0.046–0.427; *p* ≤ 0.05). The *personal norm* was the most influential psychological sub-factor (the highest β value).

## 4. Discussion

During this study, it was found that the most difficult and violated health protocol practices were reducing mobilization, followed by physical distancing, and avoiding crowds. The best possible explanation could be difficulties in reducing mobilization among young students, as they believed they were young, fit, and are at lower risk to be infected or would have mild symptoms when infected [31]. Moreover, university students have the perception that reducing mobilization might limit their social activity. The location they lived in may not support their social activities, such as having a stable internet connection in order to join a class, laboratory, library, or other facilities. Thus, students preferred to go outside the home.

Another more difficult and frequently violated health protocol was avoiding crowds. Since students are unable to reduce mobilization, it causes them to be unable to avoid crowds. On the other hand, research suggests that students enjoy being in crowds to build bonds through physiological synchrony [32]. Our finding is that physical distancing was often violated and difficult to do, which is in line with other studies [33]. Furthermore, Indonesian culture celebrates socializing in groups [34], which indeed also applies to young people. Thus, being in a crowd with friends could be a risk for the spread of the virus, which should be a concern in the COVID-19 pandemic situation.

Findings also showed that about 90% practiced handwashing, as they believed that hand washing with soap could help reduce COVID-19 transmission [35]. Equally, respondents considered wearing a mask to be easy and convenient. Students’ understanding of the importance of wearing a mask to minimize virus transmission could be the reason for these opinions [36].

Covering the mouth and nose when sneezing has the highest mean score in the daily health practice category. The possible explanation for this could be the level of education of the respondents. Research showed that the majority of highly educated respondents (graduate with a diploma or above) already understand that viruses can be transmitted through coughing and sneezing [37], which may result in the covering of the mouth and nose while sneezing. Additionally, the campaign of cough etiquette by the Ministry of Health before the pandemic could have had a positive impact on students’ behavior regarding sneezing [38].

Physical exercise has the lowest mean score compared to other daily health practices. Another study found that the frequency of physical exercise among students decreased during the COVID-19 pandemic, e.g., in Europe and Northern America [39] and also in Indonesia [40]. Therefore, relevant health promotion is needed so that students can be physically active during the pandemic in order to increase their healthy immune system.

The high score of Cronbach’s α of PCA of the three and four health protocol variables, except mobilization, indicated that those variables were found to be significantly correlated, such as regular hand washing in public places, maintaining physical distancing in a crowd, and proper mask use. This finding implies that the younger generation thinks that the four health protocols are “a unit”, i.e., they have to do all four practices. However, this high correlation does not apply to the variable of mobilization. Bivariate tests show that there was no significant correlation between variable mobilization with the other four variables.

The mean for variable mobilization was the lowest compared to the other four health protocols indicating that students still travel despite the COVID-19 situation following health protocols. At the time of data collection, the Indonesian government tried to minimize mobilization by requiring people to take PCR tests before traveling. It was claimed by the government that the PCR tests helped to reduce domestic mobilization, especially during the period of increased cases in July 2021 [41].

This study also revealed that the *personal norm (norm)* was the most influential sub-factor in the regression analysis, meaning that if the student thinks they are personally obliged to practice health protocol, they will practice it. This personal norm may contradict other norms, i.e., descriptive and injunctive norms [20]. For example, they may feel the responsibility to keep wearing a mask in a public space even though their companions do not use it. Personal norms were found to be critical in the context of environmental behavior compared to other types of norms, e.g., social or descriptive norms [42].

The personal norm should be the main target of the behavioral change intervention among university students. There are some potential behavioral change interventions, according to RANAS [27]. First, students could be elucidated about regrets when not following or practicing health protocols, including the consequences to their close family, i.e., they could be a virus carrier. Second, informing students that they should follow instructions and become a good example for others, e.g., relatives and children, etc., by practicing the health protocols. All these examples of interventions can increase a person’s sense of ethical self-obligation to practice health protocols correctly and regularly. By doing this, it can be expected that during periods of on-site or offline learning, the chance of virus transmission could be minimized.

The next influential psychological factors in the regression analysis were *belief about time (attitude)* and *action control (self-regulation).* A previous study that was conducted in the pandemic situation indicated a significant relationship between attitudes about COVID-19 and health protocols [43]. Attitude is crucial because it influences behavior and actions, e.g., *action control*. This finding indicates that the chance of performing health protocols increases if university students have the perception that health protocols do not consume time and always pay attention to the resources needed to practice health protocols.

There are some limitations of this study. The absence of many answers to the question related to avoiding crowded places limits the detailed understanding of the health protocols, i.e., only 78 respondents provided this information. However, a high Cronbach alpha value using four health protocol practices, including avoiding crowded places, indicates that students avoid crowded places as much as they practice handwashing, physical distancing, and mask use. The health protocol practices were self-reported and therefore were subjected to recall bias, especially for practices before the pandemic (before January 2020) and in April 2021. Future studies can be focused on the effect of different types of promotional activities, communication channels, and/or promoters on the health protocol practices among students. This can help to design appropriate behavioral change interventions in the future, not only in the case of COVID-19 but also in other types of emergency situations. Contextual factors, e.g., facilities to practice handwashing, university policy, etc., is another important aspect that one should also include in future studies. That is because these factors may influence perceptions and behavior [44]. Moreover, 292 respondents may not be sufficient to represent more than 1 million students in the three cities. The online survey’s link was distributed online to our networks in three cities and participation was voluntary. Therefore, we only attained 292 respondents. However, this number is still sufficient to perform statistical analysis, i.e., a minimum of 10–15 respondents per independent variable [45]. Furthermore, female students dominated our respondents, which may create a gender bias in the results. In general, the number of female university students in Indonesia is higher than male students, i.e., the ratio of females to males is 1.3:1 [46]. Another COVID-19-related study in Indonesia also had a higher number of female respondents compared to male respondents, i.e., the ratio of females to the male is 1.5:1 [9]. Moreover, since the questionnaire link was distributed to as many students or networks as possible and the participation was voluntary, we were not able to target respondents’ gender specifically. We also acknowledge that there is a chance of sampling errors, e.g., non-university students filling in the survey. The future study should think about how to minimize this in the online survey which targets a specific group, while still considering the anonymity of the respondents. Finally, this study was conducted in the Indonesian setting, which may not apply to other countries or regions, since underlying psychological factors may vary in different locations [27]. Therefore, future studies in different locations or settings are still needed to provide us with a better understanding of the health protocol practices among university students.

## 5. Conclusions

This study analyzed students’ health protocol and daily health practices in three cities in Indonesia during three periods of time: before the pandemic (pre-pandemic), during the time of decreasing daily cases (post-pandemic), and during the pandemic in 2021. The underlying psychological factors associated with the health protocols were also investigated. Comparing the mean scores in those three periods of time, this study found that the students’ health protocol and daily healthy practices increased, especially at the peak of the pandemic. The highest mean score for health protocol practices was mask use and reducing mobilization had the lowest. In terms of daily health practices, the highest mean score was covering the mouth and nose when sneezing, while physical exercise was the lowest mean score. The most often violated health protocol and most difficult to implement was reducing mobilization, followed by physical distancing. The PCA results indicate a high correlation between four health protocol practices: handwashing in the public space, physical distancing, mask use frequency, and avoiding crowded places. Moreover, three significant psychological sub-factors were positively related to students’ health protocols, from the most significant one to the least: personal norm (norm), action control (self-regulation), and belief about time (attitude).

## Figures and Tables

**Figure 1 ijerph-19-10630-f001:**
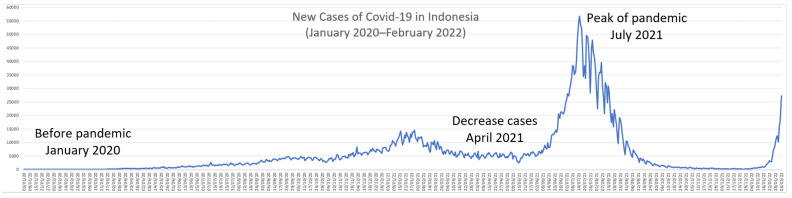
New cases of COVID-19 in Indonesia [3].

**Figure 2 ijerph-19-10630-f002:**
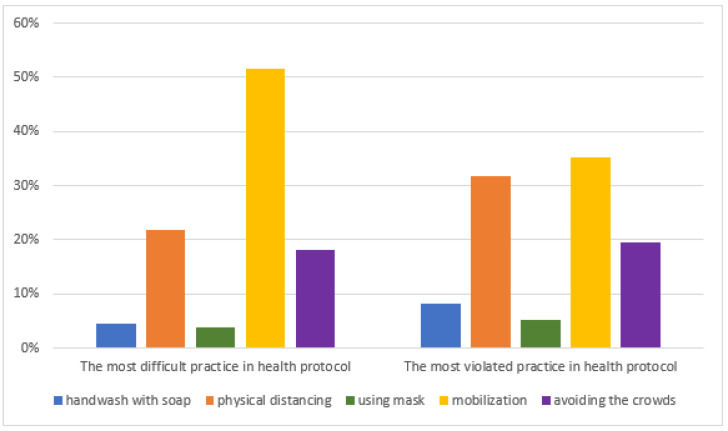
Respondent proportions regarding health protocol practices that are the most difficult to carry out and the most violated in the pandemic.

**Table 1 ijerph-19-10630-t001:** Descriptive analyses of health protocols practiced by respondents.

Psychosocial Factors		Example Question	Scale	M (SD)
**Risk**	Vulnerability	How high do you feel is the risk that you will get COVID-19?	1–5	2.47 (1.17)
	Severity on life	Imagine you get COVID-19, how severe would the impact be on your life?	1–4 *	3.94 (0.71)
	Health knowledge	Answer five true-false questions related to COVID-19	1–5	4.09 (0.88)
**Attitude**	Health benefit	How certain are you that health protocols can prevent you from getting COVID-19?	1–5	3.92 (1.11)
	Belief about time	Do you think that health protocols consume a lot of time?	1–5	4.26 (1.03)
	Belief about effort	Do you need a lot of effort to carry out health protocols?	1–5	3.48 (1.06)
	Affective belief	How much do you enjoy carrying out health protocols?	1–5	3.52 (0.89)
**Norm**	Descriptive	How many people around you always practice health protocols?	1–5	3.31 (1.11)
	Injunctive	How do people who are important to you think you should practice health protocols?	1–5	4.33 (0.88)
	Personal	How strongly do you feel an obligation to yourself to always practice health protocols?	1–5	4.6 (0.67)
**Ability**	Self-efficacy	How certain are you that you can always practice health protocols?	1–5	4.13 (0.74)
	Maintenance self-efficacy	How certain are you that you can always practice health protocols even though people around you do not practice it?	1–5	4.1 (0.8)
	Recovery self-efficacy	Imagine that you do not practice health protocols for several days, how sure are you that you will practice health protocols again?	1–5	3.55 (0.73)
**Self-regulation**	Action planning	Do you have a plan in mind for how you will practice health protocols when outside the home?	1–0 *	0.67 (0.47)
	Action control	How much do you pay attention to the resources needed to practice health protocols?	1–5	4.36 (0.73)
	Coping planning	Do you have a plan in mind for how you will practice health protocols even though people around you do not practice them?	1–0 *	0.64 (0.48)
	Commitment	How important is it for you to practice health protocol?	1–5	3.42 (1.19)

* For *health knowledge*, the scale is based on the correct items mentioned by the respondents; for *action* and *coping planning*, 1 = has a clear solution, 0 = no clear solution.

**Table 2 ijerph-19-10630-t002:** Descriptive analyses of health protocol practiced by respondents.

Variables	Mean (SD)		
	Before Pandemic	Decrease Cases	Increase Cases
Health protocol practices			
Handwashing in public spaces	3.66 (0.92)	4.23 (0.91)	4.43 (0.87)
Physical distancing	2.92 (1.19)	3.95 (0.80)	4.26 (0.81)
Using masks	3.40 (1.35)	4.66 (0.73)	4.75 (0.66)
Mobilization	2.15 (1.40)	2.15 (1.13)	2.19 (0.97)
Avoiding crowds	3.04 (1.15)	3.99 (0.66)	4.37 (0.94)
Daily healthy practices			
Covering mouth and nose when sneezing	4.31 (0.83)	4.63 (0.62)	4.68 (0.63)
Physical exercise	1.78 (0.90)	1.88 (0.96)	1.93 (1.02)
Consumption of vitamins or supplement	2.38 (1.36)	2.96 (1.33)	3.46 (1.36)
Consumption of balanced nutrition	3.67 (1.28)	3.87 (1.28)	4.01 (1.27)

Notes: range scale for handwash in public space, distancing, using mask, crowds, covering mouth and nose when sneezing, consumption vitamin or supplement, and the consumption of balanced nutrition using Likert scales: (a) 1–5, higher scores mean better practices and (b) mobilization and physical exercise were normalized so that the highest score was 5.

**Table 3 ijerph-19-10630-t003:** Regression analysis of all RANAS psychological sub-factors on HWT practice. Control variables, i.e., cities, were also included in the regression.

Variables	B	SE B	β	95% CI
*Control variables*				
Yogyakarta	0.131	0.111	0.065	−0.087–0.35
Surakarta	−0.032	0.159	−0.011	−0.346–0.282
*Risk*				
Vulnerability	−0.050	0.043	−0.059	−0.135–0.034
Severity to life	−0.077	0.069	−0.054	−0.213–0.06
Health knowledge	0.072	0.057	0.063	−0.041–0.184
*Attitude*				
Health benefit	0.041	0.048	0.046	−0.054–0.136
Belief about time	0.116	0.055	0.119 *	0.008–0.224
Belief about effort	−0.038	0.049	−0.040	−0.134–0.058
Affective belief	−0.004	0.060	−0.004	−0.122–0.113
*Norm*				
Descriptive	0.089	0.049	0.098	−0.009–0.186
Injunctive	−0.003	0.065	−0.002	−0.13–0.125
Personal norm	0.344	0.099	0.232 **	0.149–0.539
*Ability*				
Self-efficacy	0.103	0.113	0.077	−0.118–0.325
Maintenance self-efficacy	−0.036	0.100	−0.029	−0.234–0.161
Recovery self-efficacy	0.136	0.074	0.099	−0.01–0.281
*Self-regulation*				
Action planning	0.183	0.119	0.086	−0.051–0.417
Action control	0.236	0.097	0.173 *	0.046–0.427
Coping planning	−0.131	0.116	−0.063	−0.36–0.097
Commitment	0.051	0.047	0.060	−0.041–0.143

* *p* ≤ 0.05, ** *p* ≤ 0.01. Adjusted R^2^ = 0.353, n = 292.

## Data Availability

Not applicable.

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
