# Peer review of "The COVID-19 Health Protocol among University Students: Case Studies in Three Cities in Indonesia"

_ijerph, 2022, doi:10.3390/ijerph191710630_

Round 1
Reviewer 1 Report
Line 38 –Experience ( GET)
Line 56 Edit (SOME)
Line 95 ( S/B Practices (?) more than 1 . There is ref to this throughout paper , may be helpful .
Line 106( Describe the google form)
Line 120 ( Great descriptions RANAS , but spell out full acronym when introducing in heading )
Line 144( Very descriptive Table 1, using PSY factors)
Line 223( Informative and interconnected discussion points) Excellent analysis.
Is there any insight as to high percentage of female VS male respondents.
May be interesting to replicate study or determine is similar research found gender population sample unique.
Author Response
Line 38 –Experience (GET)
Response: Thank you for the revision. The word “get” has been replaced by “experience”.
People who are infected with COVID-19 will experience mild to moderate symptoms however, those who have the underlying medical condition are likely to get severe ill-ness, which can lead to death. --Line 40-42
Line 56 Edit (SOME)
Response: We replaced the word “some” with “several”.
Those studies found several significant psychological factors related to the practice of health protocol, e.g., the social norm in the community, injunctive norms, intention, and perceived self‑efficacy. --Line 57-60
Line 95 (S/B Practices (?) more than 1. There is ref to this throughout paper, may be helpful.
Response: We have changed it to “practices”. We have also provided citations to sentences that need a citation.
This may help us understand the dynamics of health protocol practices among university students. --- line 96-97
Line 106 (Describe the google form)
Response: We elaborated further in the section 2.1.
Google form is one of some instruments of google workspace. It is used for create online questionnaire that easy to be published and analyzed in the real time [25]. The most important benefit of using Google Form is easily obtaining people’s opinion remotely [26]. In this research, we created some questions about respondent’s background (number of questions=8), implementation of health protocol (n=11), and RANAS {Risks (n=7), Attitudes (n=7), Norms (n=3), Abilities (n=3), and Self‐regulation (n=7)}. --Line 114-119
Line 120 (Great descriptions RANAS, but spell out full acronym when introducing in heading)
Response: We have written the spell out of RANAS in the heading section.
2.2. RANAS (Risk, Attitude, Norm, Ability, and Self-regulation) Psychological framework --Line 127
Line 144 (Very descriptive Table 1, using PSY factors)
Response: Table 1 is for describing in detail what questions we used in the survey.
Line 223 (Informative and interconnected discussion points) Excellent analysis.
Response: Thank you for your comment.
Is there any insight as to high percentage of female VS male respondents.
Response: We added extra explanation about this in the discussion section.
Finally, female students dominated our respondents which may create a gender bias in the results. In general, the number of female university students in Indonesia is higher than male students, i.e., ratio female: male is 1.3:1 [47]. Other COVID-19-related study in Indonesia also had the higher number of female respondents compared to male respondents, i.e., ratio female: male is 1.5:1 [10]. Moreover, since the questionnaire link was distributed to as many students or networks as possible and the participation was voluntary, we cannot target specific respondent’s gender. --Line 324-331
May be interesting to replicate study or determine is similar research found gender population sample unique.
Response: Please see our previous response regarding this.
Reviewer 2 Report
Explain RANAS earlier, where the authors used the abbreviated first time.
The authors discussed various psychological theories of health promotion in the introduction but did not explain which theory they employed to carry out this study.
Data collection:
In the abstract, it is mentioned that three districts were selected randomly. However, in the method section line (109-113) it is mentioned that these three districts were selected due to having many universities. This seems purposive sampling instead of random sampling.
In abstract and method section, it is mentioned that the data were collected in 3-time phases (before the pandemic, decrease cases, and the peak of the pandemic). Is there any description related to how many cases were collected before the pandemic, decrease cases, and the peak of the pandemic?
Moreover, if the authors collected the data in three time zones, then I could not see the sample size of these three time zones in the method section.
I would also suggest analyzing/presenting the descriptive data on three time zones clearly to give key messages.
Also clearly mention the time zone when the data were collected in three time zones in the method section.
Did authors develop their own scale or used the pre-designed scale of previous studies?
What is the reliability score of this tool? Did the authors pre-tested this tool?
Regression analayis:
It is mentioned that the regression analysis is shown in table two. However, the headline of table 2 is "Table 2. Descriptive analyses of health protocol practiced by respondents". Where is the regression analysis table?
The discussion section has a lack of comparison of the results of the data collected in three time zones. I would suggest writing the discussion section on comparision of the data collected in three time zones.
Author Response
Explain RANAS earlier, where the authors used the abbreviated first time.
Response: We already write down the spell out of RANAS.
Some psychological frameworks or theories have been used to analyze the practice of health protocol, e.g., the health belief model, the reasoned action approach, and the risk, attitude, norms, ability, and self-regulation (RANAS) framework [8–10]. --Line 55-57
The authors discussed various psychological theories of health promotion in the introduction but did not explain which theory they employed to carry out this study.
Response: Some psychological theories mentioned in the introduction are to tell readers that previous studies have utilized some theories in explaining COVID-19-related behaviour. However, in this study, we used RANAS.
RANAS psychological framework was used to understand underlying psychological factors of the COVID-19 health protocol practice. RANAS framework consists of psychological factors related to risk, attitude, norm, ability, and self-regulation that potentially affect personal behavior [10, 11]. --Line 91-94
Previous studies also used RANAS framework model to find critical psychological factors that influence the health protocols, for example, public perception and hand hygiene behavior [10] and mask-wearing [12]. -- Line 136-138
Data collection:
In the abstract, it is mentioned that three districts were selected randomly. However, in the method section line (109-113) it is mentioned that these three districts were selected due to having many universities. This seems purposive sampling instead of random sampling.
Response: Thank you for your correction. We have edited the abstract and also introduction.
The designed online survey was conducted among 292 university students in three cities in Indonesia, i.e., Yogyakarta, Semarang, and Surakarta. -- Line 20-21
University students from three cities in Indonesia were considered for this study: Yogyakarta, Semarang, and Surakarta. These areas have more than 130 universities and a large number of students. -- Line 84-87
In abstract and method section, it is mentioned that the data were collected in 3-time phases (before the pandemic, decrease cases, and the peak of the pandemic). Is there any description related to how many cases were collected before the pandemic, decrease cases, and the peak of the pandemic?
Response: The number of cases is shown in the figure 1. We updated the information about the estimated time for each phase.
The purpose of this study was to get an overview of changes in university students' behaviour in implementing health protocol (wearing masks, maintaining distance, washing hands, staying away from crowds, and limiting mobility) related to the COVID-19 pandemic in 3-time phases (before the pandemic, i.e., before January 2020, decrease cases, i.e., around April 2021, and the peak of the pandemic, i.e., July 2021) that are shown in figure 1.
Figure 1. New cases of COVID-19 in Indonesia [3]. -- Line 102-109
Moreover, if the authors collected the data in three time zones, then I could not see the sample size of these three time zones in the method section.
Response: The data collection was not conducted in each the three phases, but during July 2021-February 2022. However, we asked respondents to estimate or recall their past behaviour in the first and second phases. We added extra information about this in the method to avoid confusion. Limitation of this method is also presented in the discussion.
It is important to note that the data collection was conducted only during July 2021-February 2022, but we asked respondents to recall their behavior for the first phase, i.e., before pandemic, and the second phase, i.e., decrease case. -- Line 111-113
The health protocol practices were self-reported and therefore were subjected to recall bias, especially for practices before the pandemic (before January 2020) and in April 2021. -- Line 312-314
I would also suggest analyzing/presenting the descriptive data on three time zones clearly to give key messages.
Response: Please see our previous response regarding this.
Also clearly mention the time zone when the data were collected in three time zones in the method section.
Response: Please see our previous response regarding this.
Did authors develop their own scale or used the pre-designed scale of previous studies?
Response: We developed our own scales for variables related to health protocol, such as mobilization and physical exercise. However, for the RANAS psychological variables, we follow the scale suggested by the RANAS guidelines.
The measurement of the frequency of health protocols and daily healthy practices using Likert scales, which divided further into five scales: never, very rarely, seldom, almost, and always, except reducing mobilization, i.e., range score was 1 (best practices) – 10 (worst practices), and physical exercise, i.e., range score was 1 (no physical exercise at all) - 9 (seven days in a week). -- Line 149-153
All scales used in the RANAS-related questions follow the RANAS guidelines [28]. -- Line 135-136
What is the reliability score of this tool? Did the authors pre-tested this tool?
Response: The reliability analysis of the health protocol-related questions was conducted, especially because we combined them using the principal component analysis. We set the minimum value of the Cronbach’s α of 0.7 as a threshold. The results are also presented.
The PCA procedure was deemed acceptable when the minimum Kaiser-Meyer-Olkin (KMO) value was 0.5 [30] and the Cronbach’s α value was above 0.7 [31]. -- Line 165-166
The KMO value was 0.676, which can explain 64% of the variance (n= 292). Furthermore, the Cronbach’s α value was 0.714. PCA by adding an extra health protocol practice, i.e., avoiding crowded places was also included. The KMO and Cronbach’s α values were 0.723 and 0.725, respectively, which can explain 57% of the variance (n= 78). -- Line 212-216
However, we did not do the reliability analysis for RANAS-related questions because we basically follow all RANAS sub-factor questions mentioned in the RANAS guidelines. The questionnaire is pre-tested with some university students before distributed.
Regression analysis:
It is mentioned that the regression analysis is shown in table two. However, the headline of table 2 is "Table 2. Descriptive analyses of health protocol practiced by respondents". Where is the regression analysis table?
Response: Thank you for the correction. Table 2 is missing. Our apologies for this. We have revised it.
Table 3. Regression analysis of all RANAS sub-factors psychological factors on HWT practice. Control variables, i.e., cities, were also included in the regression.
|
Variables |
B |
SE B |
β |
|
Control variables |
|
|
|
|
Yogyakarta |
0.131 |
0.111 |
0.065 |
|
Surakarta |
-0.032 |
0.159 |
-0.011 |
|
Risk |
|||
|
Vulnerability |
-0.050 |
0.043 |
-0.059 |
|
Severity on life |
-0.077 |
0.069 |
-0.054 |
|
Health knowledge |
0.072 |
0.057 |
0.063 |
|
Attitude |
|||
|
Health benefit |
0.041 |
0.048 |
0.046 |
|
Belief about time |
0.116 |
0.055 |
0.119 * |
|
Belief about effort |
-0.038 |
0.049 |
-0.040 |
|
Affective belief |
-0.004 |
0.060 |
-0.004 |
|
Norm |
|||
|
Descriptive |
0.089 |
0.049 |
0.098 |
|
Injunctive |
-0.003 |
0.065 |
-0.002 |
|
Personal norm |
0.344 |
0.099 |
0.232 ** |
|
Ability |
|||
|
Self-efficacy |
0.103 |
0.113 |
0.077 |
|
Maintenance self-efficacy |
-0.036 |
0.100 |
-0.029 |
|
Recovery self-efficacy |
0.136 |
0.074 |
0.099 |
|
Self-regulation |
|||
|
Action planning |
0.183 |
0.119 |
0.086 |
|
Action control |
0.236 |
0.097 |
0.173* |
|
Coping planning |
-0.131 |
0.116 |
-0.063 |
|
Commitment |
0.051 |
0.047 |
0.060 |
*p ≤ 0.05, **p ≤ 0.01, ***p ≤ 0.001. Adjusted R2 = 0.353, n = 292.
--Line 231-233
Reviewer 3 Report
This study examines the differences in health protocol and health practices among university students in three phases of the COVID-19 pandemic (pre-pandemic, during the pandemic, and post-pandemic) using an online survey.
The subject of the article is interesting; however, the statistical analysis must be improved.
The authors didn’t justify why a sample with 292 respondents (of which 74% are female) in a population of over 1 million students is representative. The sample error, confidence interval are also missing.
The methods used were not briefly described in the Abstract and Introduction.
It´s not clear the practical and scientific contribution of the paper.
The authors also didn´t follow IJERPH format guidelines. References are not numbered in order of appearance in the text.
Author Response
This study examines the differences in health protocol and health practices among university students in three phases of the COVID-19 pandemic (pre-pandemic, during the pandemic, and post-pandemic) using an online survey. The subject of the article is interesting; however, the statistical analysis must be improved.
Response: Thank you for your feedback. We have updated the draft.
The authors didn’t justify why a sample with 292 respondents (of which 74% are female) in a population of over 1 million students is representative. The sample error, confidence interval are also missing.
Response: We mention 1.1 million students in the three cities in the introduction to emphasize that there are many students in these three cities. However, we admit that 292 respondents may not be a good representative for 1 million students in the three cities. We state this as one of the limitations. However, we consider that this number is sufficient to perform statistical analysis. We also update extra information about the female respondents.
Moreover, 292 respondents may not be sufficient to represent more than 1 million students in the three cities. The online survey's link was distributed online to our networks in three cities and participation was voluntary. Therefore, we got only 292 respondents. However, this number is still sufficient to perform statistical analysis [46]. --Line 320-324
Finally, female students dominated our respondents which may create a gender bias in the results. In general, the number of female university students in Indonesia is higher than male students, i.e., ratio female: male is 1.3:1 [47]. Other COVID-19-related study in Indonesia also had the higher number of female respondents compared to male respondents, i.e., ratio female: male is 1.5:1 [10]. Moreover, since the questionnaire link was distributed to as many students or networks as possible and the participation was voluntary, we cannot target specific respondent’s gender. --Line 324-331
The methods used were not briefly described in the Abstract and Introduction.
Response: We have updated the information about methods in the abstract.
A forced-entry multivariate regression was conducted using all RANAS sub-factors as independent variables and health protocol obtained from PCA as the dependent variable. --Line 21-23
It´s not clear the practical and scientific contribution of the paper.
Response: We have discussed about the scientific contribution of this paper in the introduction.
Very limited studies related to the COVID-19 and university students have been conducted. Among these limited studies, Sharma et al., 2021 [15] considered only hand washing practice among other important health protocols for their Multi-Theory Model (MTM). However, other health protocols and practices among university students could provide important insight to prevent or spread COVID-19 within different age groups and/or communities. Therefore, this study aims to identify and fill the knowledge gap about the COVID-19 health protocol among university students. -- Line 78-84
This may help us understand the dynamics of health protocol practice among university students. Discussion and strategies to prevent COVID-19 transmission among students are also offered. --Line 96-99
Practical recommendation/contribution of this paper is presented in the discussion.
The personal norm should be the main target of the behavioral change intervention among university students. There are some potential behavioral change interventions according to RANAS [28]. First, it could explain to the students about regretting following or performing health protocols, including the consequence on their close family, i.e., they can be the virus carrier to their family. Second, informing students to follow the instructions and becoming a good example for others, e.g., relatives, children, etc., in practicing the health protocols. All these examples of intervention can create a person’s sense of self-ethical obligation to practice health protocols correctly and regularly. By doing this, it can be expected that, in the time of on-site or offline learning, the chance of virus transmission could be minimized. -- Line 290-299
The authors also didn´t follow IJERPH format guidelines. References are not numbered in order of appearance in the text.
Response: We have updated the references format according to IJERPH format guidelines. Thank you for your correction.
Round 2
Reviewer 2 Report
Thank you for sharing this paper again for review.
The authors have addressed all the comments carefully.
I still have a minor reservation about collecting the data on three different time zones on recall biases. That creates a bit of confusion while correlating the results. However, the authors have also mentioned this in the conclusions/limitation of the study.
Author Response
Thank you for sharing this paper again for review.
The authors have addressed all the comments carefully.
Response: Thank you for your comment in the first round.
I still have a minor reservation about collecting the data on three different time zones on recall biases. That creates a bit of confusion while correlating the results. However, the authors have also mentioned this in the conclusions/limitation of the study.
Response: We agree that there is a chance for recall bias regarding the first and second time zones, as stated in the limitation. However, our statistical analyses, i.e., PCA and regression, are mainly focus on the third time zone, when the questionnaire was distributed. Furthermore, to avoid confusion, we decide to re-construct our story line and focus more on the third time zone. We deleted and edited some sentences in the abstract and also introduction.
The changes in the health protocols practice were also investigated, i.e., before the COVID-19 entered Indonesia when the cases were down, and the peak phase of the pandemic in 2021, which was caused by the Delta variant [21], even though this study is more focuses on the peak phase, i.e., when the data was collected. --- line 93-96
Reviewer 3 Report
Although the authors modified substantially the main text, some statistical aspects of the paper still must be improved.
Please, in the Introduction frame the questions being addressed, describe the methods briefly and provide context for the findings being presented.
The state of the art and the practical and scientific contribution of the paper is vague.
In line 114, the authors state “Very limited studies” (in plural) but didn´t explain why they are limited. They just present 1 paper (Sharma et al., 2021).
The Introduction is confusing. Please, review the sequence of paragraphs, and clarify the research question and the academic/practical contribution of the paper, making the introduction comprehensible to scientists and the public in general.
It´s not clear if the paper is useful only for Indonesia.
Line 458: Why 292 respondents is sufficient to perform the statistical analysis?
The sample error and confidence interval are also still missing, or the absence of this topic is not scientifically justified. The authors just put references [46, 47].
Author Response
Although the authors modified substantially the main text, some statistical aspects of the paper still must be improved.
Response: Thank you for your comment. We have improved the draft following your suggestion.
Please, in the Introduction frame the questions being addressed, describe the methods briefly and provide context for the findings being presented.
Response: We have updated the introduction with research questions, brief methodology, and finding’s context.
Therefore, this study aims to identify and fill the knowledge gap about the COVID-19 health protocol among university students. More specifically, the research questions are: (1) “How is the practice of health protocols among university students in Indonesia?”, and (2) “What are the psychological factors that influence those health protocols?”. --- line 83-87
The online survey data of university students from three cities in Indonesia were collected for this study: Yogyakarta, Semarang, and Surakarta. RANAS psychological framework was followed to understand underlying psychological factors of the COVID-19 health protocol practice. --- line 88-91
The data were analyzed statistically, especially to find out significant psychological factors associated with the health protocols. Understanding underlying psychological factors behind the behavior is necessary to design relevant strategies to prevent COVID-19 transmission among university students. --- line 97-101
The state of the art and the practical and scientific contribution of the paper is vague.
Response: We have updated the knowledge gap and research questions of this study. We hope that it can make clear the scientific contribution of this study.
However, a very limited study analyzes the underlying psychological factors related to the COVID-19 health protocols among university students. For example, Sharma et al. [13] use Multi-Theory Model (MTM) to analyze health protocols in students, but only for the hand washing practice. However, we argue that analyzing other health protocols and practices among university students, including underlying psychological factors, could provide important insight to prevent or spread of COVID-19 within different age groups and/or communities. Therefore, this study aims to identify and fill the knowledge gap about the COVID-19 health protocol among university students. More specifically, the research questions are: (1) “How is the practice of health protocols among university students in Indonesia?”, and (2) “What are the psychological factors that influence those health protocols?”. --- line 77-87
Understanding underlying psychological factors behind the behavior is necessary to design relevant strategies to prevent COVID-19 transmission among university students. --- line 99-101
In line 114, the authors state “Very limited studies” (in plural) but didn´t explain why they are limited. They just present 1 paper (Sharma et al., 2021).
Response: We have updated the text to avoid confusion.
However, a very limited study analyzes the underlying psychological factors related to the COVID-19 health protocols among university students. For example, Sharma et al. [13] use Multi-Theory Model (MTM) to analyze health protocols in students, but only for the hand washing practice. However, we argue that analyzing other health protocols and practices among university students, including psychological factors, could provide important insight to prevent or spread of COVID-19 within different age groups and/or communities. --- line 77-83
The Introduction is confusing. Please, review the sequence of paragraphs, and clarify the research question and the academic/practical contribution of the paper, making the introduction comprehensible to scientists and the public in general.
Response: We have re-arranged the introduction, updated the research questions, and scientific contribution. Please see the previous responses to this topic.
It´s not clear if the paper is useful only for Indonesia.
Response: Significant psychological factors influencing the behavior may vary depending on the location or settings. Thus, this study is more useful for the Indonesian context. We have stated this as one of the study limitations.
Finally, this study was conducted in the Indonesian setting, which may not apply to other countries or settings, since underlying psychological factors may vary in different locations [27]. Therefore, future studies in different locations or settings are still needed to provide us with a better understanding of the health protocol practices among university students. --- line 340-344
Line 458: Why 292 respondents is sufficient to perform the statistical analysis?
Response: We edited the text.
Therefore, we got only 292 respondents. However, this number is still sufficient to perform statistical analysis, i.e., a minimum of 10-15 respondents per independent variable [45]. --- line 331-333
The sample error and confidence interval are also still missing, or the absence of this topic is not scientifically justified. The authors just put references [46, 47].
Response: We have added the confidence interval in table 3. We acknowledge that there is a chance of sampling error. We have added this in the limitation.
Moreover, since the questionnaire link was distributed to as many students or networks as possible and the participation was voluntary, we cannot target specific respondent’ gender. We also acknowledge that there is a chance of sampling error, i.e., non-university students fill in the survey. The future study should think about how to minimize this in the online survey which targets a specific group, while still considering the anonymity of the respondents. --- line 338-343